# Question Difficulty Consistent Knowledge Tracing

## ABSTRACT

Knowledge tracing aims to estimate knowledge states of students based on their historical learning activities. Many deep learning models have been developed for knowledge tracing with impressive performance. Early works like DKT use skill IDs and student responses only. Recent works also incorporate questions IDs into their models and achieve much improved performance. However, predictions made by these models are thus on specific questions, and it is not straightforward to translate them to estimation of students' knowledge states over skills. In this paper, we propose to replace question IDs with question difficulty levels in deep knowledge tracing models, which transforms the knowledge tracing problem to "predicting whether a student can answer any question of a given skill at a given difficulty level correctly". The predictions made by our model can be more readily translated to students' knowledge states over skills. Furthermore, by using question difficulty levels to replace question IDs, we can also alleviate the cold-start problem in knowledge tracing as online learning platforms are updated frequently with new questions. We further use two techniques to smooth the predicted scores. One is to combine embeddings of nearby difficulty levels using a Hann function. The other is to constrain the predicted probabilities to be consistent with question difficulty levels by imposing a penalty if they are not consistent. We conduct extensive experiments to study the performance of the proposed model. Our experiment results show that our model outperforms latest knowledge tracing models in terms of both AUC/RMSE and consistency with question difficulty levels.

## CCS CONCEPTS

• **Computing methodologies** → **Machine learning algorithms**;
• **Applied computing** → *Education*.

## KEYWORDS

Knowledge tracing, deep sequence models, cold-start questions

**ACM Reference Format:**
Anonymous Author(s). 2018. Question Difficulty Consistent Knowledge Tracing. In *Proceedings of Make sure to enter the correct conference title from your rights confirmation emai (Conference acronym 'XX)*. ACM, New York, NY, USA, 9 pages. https://doi.org/XXXXXXX.XXXXXXX

## 1 INTRODUCTION

Knowledge tracing aims to estimate knowledge states of students over a set of knowledge components (also called skills) based on

their historical learning activities. It enables the possibility of providing personalized feedbacks and recommendations in a timely manner in online education. It is a key component in intelligent tutoring systems (ITSs). Let $x = (u, q, y)$ be a learning activity of a student, where $u$ is the student ID, $q$ is a question ID, and $y$ is a binary variable (class label) indicating whether student $u$ answered question $q$ correctly or not. Each question has one or more skills associated with it. The knowledge tracing problem can be formulated as follows: given a sequence $S_u = \langle x_1, x_2, \cdots, x_t \rangle$ containing historical learning activities of a student $u$, predict whether student $u$ can answer the next question correctly.

Many approaches have been proposed to tackle the knowledge tracing problem, including traditional Bayesian Knowledge Tracing (BKT) [9], factor analysis [3, 33], recurrent neural networks [20, 27, 34, 47] and attentive models [8, 11, 29, 38]. Deep learning based models have shown superior performance over traditional methods. Early deep learning based models like DKT [34] use skill IDs and student responses only. More recent works [11, 38] also use questions IDs and achieve much improved performance. However, predictions made by these models are thus on specific questions, and it is not straightforward to translate them to students' knowledge states over skills.

In this paper, we propose to use question difficulty levels to replace question IDs in deep knowledge tracing models. By doing so, we transform the knowledge tracing problem from *predicting whether a student can answer the next specific question of a given skill correctly* to *predicting whether a student can answer any question of a given skill at a given difficulty level correctly*. The predictions made by our model can be more easily used to estimate students' knowledge states over skills. Furthermore, by using question difficulty levels to replace question IDs, we can also alleviate the cold-start problem in knowledge tracing as online learning platforms are updated frequently with new questions. Knowledge tracing models that rely on question IDs may not perform well on new questions. Difficulty levels of questions can be obtained using different methods, such as being estimated from learning activity data, annotated by domain experts, or estimated from contents of questions using language models.

We adopt two techniques to further smooth predictions made by our model. One is to combine the embedding of a difficulty level with the embeddings of its neighbors using a Hann function. The other one is to constrain the predicted scores to be consistent with questions difficulty levels. More specifically, given a student, a question with a higher difficulty level should have a lower probability of being answered correctly by the student than a question on the same skill with a lower difficulty level. We impose a penalty if this constraint is not satisfied.

The main contributions of this paper can be summarized as follows:

- We use question difficulty levels to replace question IDs in our knowledge tracing model. The predictions made by our

model can be more readily translated to students' knowledge states over skills than models that rely on question IDs. It also eases the cold-start problem in knowledge tracing.

- We further use two techniques, combining embeddings of nearby question difficulty levels and question difficulty consistent constraint, to produce predictions that are more aligned with difficulty levels of questions, which make it easier for end users to use and trust the predictions.
- We propose a simple and efficient model architecture which uses a LSTM sublayer to learn representations of historical learning sequences and a feed-forward neural network (FFN) as the prediction layer. Despite its simplicity, our model outperforms latest deep learning based knowledge tracing models that use more complex architectures.
- We conduct extensive experiments to study the performance of our models. Our experiment results show that our model indeed produces predictions that are more consistent with question difficulty levels than existing models, and it is also more efficient and more accurate.

The rest of the paper is organized as follows. Section 2 introduces related work. Section 3 presents our question difficulty consistent knowledge tracing model. Experiment results are reported in Section 4. Finally, Section 5 summarizes and concludes the paper.

## 2 RELATED WORK

In this section, we briefly introduce the different approaches for knowledge tracing. For a comprehensive review of these algorithms, please refer to [2, 21].

Bayesian Knowledge Tracing (BKT) [9] models knowledge states of students using a Hidden Markov Model with four parameters: prior knowledge, learning rate, slip probability and guess probability. Compared with deep learning based approaches, BKT falls short in capturing inter-skill similarity, contextualized trial sequence and variation in student ability [15]. Several improvements have been made to standard BKT. [10] builds a machine learning model to estimate contextual slip and guess probabilities based on the learning sequence. [32] introduces question level difficulty to BKT by giving each question its own guess and slip probability. [31, 49] include student-specific parameters for more personalized knowledge tracing. [15] incorporates forgetting, skill grouping and latent student ability into BKT and achieves significant performance gains over standard BKT.

Another approach to knowledge tracing uses factor analysis based on Item Response Theory (IRT). Learning factor analysis (LFA) [3, 4] has parameters on student prior knowledge, knowledge learning rate and question difficulty level but it assumes students learn at the same rate. Performance factor analysis (PFA) [33] further improves LFA by considering number of previous correct answers and wrong answers. DASH [19] uses multiple time windows to capture effects of learning and forgetting over time. All these three algorithms cannot capture connections among knowledge components. Sparse factor analysis (SPARFA) [17] uses matrix factorization to predict students' performance on a set of questions. KTM [42] use Factorization Machines to incorporate side information for knowledge tracing. Both SPARFA and KTM can capture connections among questions via latent factors but they ignore the sequential order of student-question interactions. SPARFA-Trace [16] extends SPARFA by further modeling knowledge state transitions that are induced by learning and forgetting over time. DAS3H [7] combines DASH and KTM to capture both connections among questions/skills and forgetting behaviors.

With the success of deep learning in various domains, many deep learning models have been applied to knowledge tracing, including ConvNN [37], memory-augmented NN [1, 51], Graph NN [28, 40, 46, 50], RNN [5, 20, 22, 23, 26, 34, 43, 45, 47] and attentive models [8, 11–13, 18, 29, 30, 38, 44, 48]. Deep learning based models are shown to perform better than traditional approaches. They differ not only in their model architectures, but also in what information is used and how the information is converted to model inputs. The first deep learning model DKT [34] uses skill IDs and student responses only. It encodes each combination of skills and student responses using either one-hot encoding or random vectors. Both are fixed and non-learnable. DKVMN[51] and SAKT[29] can take either skill IDs or question IDs as inputs, but not both. Both models use three unlinked embeddings to encode three states of a question/skill: answered correctly, answered wrongly and to-be-answered. This encoding method is adopted by many later models. Another popular encoding method is proposed by AKT [11], which uses the Rasch model to convert skill IDs and question IDs to model inputs. The Rasch model uses 1-dimension embeddings for questions, which may cause under-parametrization when the number of questions is not large.

A few works use question difficulty explicitly like our work. [25] builds an interpretable model using Tree-Augmented Naive Bayes Classifier (TAN) on three features: individual skill mastery, student ability and problem difficulty. DIMKT [35] uses both question ID embeddings and question difficulty embeddings together with skill ID embeddings and skill difficulty embeddings. Our experiment results show that the predictions made by DIMKT may change dramatically over question difficulty levels.

Besides question IDs, skill IDs and class labels, other information has also been used to better model student learning. DKT-forget [27] considers elapsed time from the previous question with the same skill and number of previous encounters of the same skill. SAINT+ [38] and LPKT [36] use two features, elapsed time from the previous question and response time. Textual contents of questions have been used in [13, 20, 30]. Relationships among knowledge components such as prerequisite and similarity are utilized in [6, 14, 40]. The performance of our model can also be further improved by incorporating such information. We will leave this to our future work.

## 3 QUESTION DIFFICULTY CONSISTENT KNOWLEDGE TRACING

In this section, we first introduce the overall architecture of our Question Difficulty Consistent Knowledge Tracing (QDCKT) model, and then describe the embedding layer and the question difficulty consistent constraint.

### 3.1 The overall architecture

Our model uses an LSTM sublayer to generate representations of historical learning sequences and a feed-forward neural network

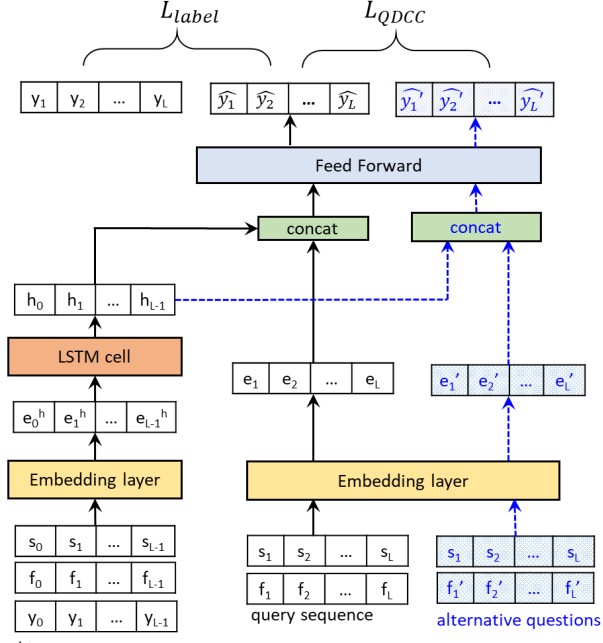

**Figure 1: Architecture of QDCKT. Length-$L$ history sequence is fed to LSTM to generate representations of history sequence. Length-$L$ query embedding sequence is concatenated with representations of history sequence and fed to FFN prediction layer to generate predicted scores. Query sequence is one position ahead of history sequence. $s_i$ is skill ID, $f_i$ is question difficulty level, and $y_i$ is class label at position $i$.**

(FFN) as the prediction layer as shown in Figure 1. For a given length-$(L + 1)$ learning activity sequence, its first length-$L$ subsequence is regarded as *history sequence* and is fed to the LSTM sublayer, and the last length-$L$ sub-sequence is regarded as *query sequence* whose class labels are to be predicted. Note that the query sequence (from 1 to $L$) is one position ahead of the history sequence (from 0 to $L - 1$), and the model is trained to predict the class labels over the whole query sequence. History sequences contain skill IDs, question difficult levels and class labels. Query sequences contain skill IDs and question difficult levels only. Class labels of query sequences are not fed to the embedding layer and they are used for calculating training loss only.

We use $q_i$ to denote question ID, $f_i$ to denote question difficulty level of $q_i$, $s_i$ to denote skill ID, $y_i$ to denote class label at position $i$ in a length-$(L + 1)$ sequence, $i$=0, 1, $\cdots$, $L$. Both the history sequence and the query sequence first pass through an embedding layer respectively to generate embeddings $e_{i-1}^h$ and $e_i$ at each position $i$, $i$=1, 2, $\cdots$, $L$. Note that skill ID embeddings and question difficulty level embeddings are shared between history sequences and query sequences. The history embedding sequence is then passed through the LSTM sublayer. For question $q_i$ at position $i$, its history sequence ends at position $i-1$. The hidden state of LSTM cell at position $i-1$, denoted as $h_{i-1}$, is regarded as the representation of the history sequence of $q_i$. It is concatenated with embedding of query sequence

at position $i$, $e_i$, and the resultant vector is passed through the FFN prediction layer to make the final prediction. The FFN prediction layer is given as below, where $W_1$, $W_2$, $b_1$, $b_2$ are learnable model parameters, and $\hat{y}_i$ is the predicted probability at position $i$, $i$=1, 2, $\cdots$, $L$.

$$\hat{y}_i = \sigma(ReLU([h_{i-1}, e_i] \cdot W_1 + b_1) \cdot W_2 + b_2) \quad (1)$$

We use both binary cross entropy loss $\mathcal{L}_{label}$ and question difficulty consistent loss $\mathcal{L}_{QDCC}$ to learn model parameters. Question difficulty consistent loss is described later in Section 3.3. Binary cross entropy loss between the ground-truth class labels $y_i$s and predicted probabilities $\hat{y}_i$s over the whole length-$L$ query sequence is calculated below.

$$\mathcal{L}_{label} = \frac{1}{L} \sum_{i=1}^{L} (-y_i \log(\hat{y}_i) - (1 - y_i) log(1 - \hat{y}_i)) \quad (2)$$

The overall loss is shown below, where $\lambda$ is a hyper-parameter.

$$\mathcal{L} = \mathcal{L}_{label} + \lambda \cdot \mathcal{L}_{QDCC} \quad (3)$$

## 3.2 Embedding layer

The inputs to our model include skill IDs, question difficulty levels and class labels. Skill IDs are mapped to $d_e$-dimensional embeddings using an embedding matrix $S \in \mathcal{R}^{M \times d_e}$, where $M$ is the number of skills. Question difficulty levels can be estimated from learning activity data, annotated by domain experts, or estimated based on question contents using pre-trained language models. Here we calculate difficulty levels of questions from learning activities, and only training data are used. Given a question $q$, let $n$ be the number of learning activities containing $q$ in training data, $n_p$ be the number of activities with positive class labels among the $n$ activities, and $p_c$ be the overall percentage of correct answers in training data. The difficulty of $q$, denoted as $diff(q)$, is calculated as below, where $\alpha$ is used for smoothing and it is set to 5 in our experiments.

$$diff(q) = 1 - \frac{n_p + \alpha \cdot p_c}{n + \alpha} \quad (4)$$

The difficulty level of $q$ is calculated by converting $diff(q)$ to a number between 1 and $N$ as follows, where $N$ is the maximal number of difficulty levels, and it is a hyper-parameter.

$$f_q = \lfloor diff(q) \cdot N \rfloor + 1 \quad (5)$$

Each difficulty level is mapped to one $d_e$-dimensional embedding using an embedding matrix $\mathcal{D} \in \mathcal{R}^{N \times d_e}$. We expect the embeddings of nearby difficulty levels are close to each other. We adopt the *ContinuousEmbedding* layer used in [41] to obtain smoothed embeddings for difficulty levels. Given a difficulty level $i$, we take the $l$ adjacent embeddings centered at $i$, and use Hann function to calculate their weighted sum to get the smoothed embedding of $i$. Here window size $l$ is a hyper-parameter. We use an example to show how this is calculated. Let $l$=5 and $i$=10. We take the five embeddings between difficulty level 8 and difficulty level 12, and denoted them as $e_j^f$, j=8, 9, $\cdots$, 12. Weights in a Hann window of 5 are $[0, 0.5, 1, 0.5, 0]$. We normalize the weights to have sum of 1, and the weights become $[0, 0.25, 0.5, 0.25, 0]$. The smoothed embedding of difficulty level 10 is then calculated as $\bar{e}_{10}^f = 0 \times e_8^f + 0.25 \times e_9^f + 0.5 \times e_{10}^f + 0.25 \times e_{11}^f + 0 \times e_{12}^f$.

Class labels are mapped to $d$-dimensional embeddings using an embedding matrix $C \in \mathcal{R}^{2 \times d}$, where $d$ is the input dimension to the LSTM sublayer and it can be different from $d_e$. To generate the input vectors to the LSTM sublayer, skill ID embeddings and question difficulty level embeddings are concatenated and then linearly transformed to $d$-dimensional vectors, and then added to the class label embeddings. More formally, let $e_i^s \in \mathcal{S}$ be the skill ID embedding, $\bar{e}_i^f$ be the smoothed question difficulty level embedding, and $e_i^y$ be the class label embedding at position $i$, $i$=0, $1, \cdots, L-1$. The input vector $e_i^h$ to LSTM is generated as follows, where $W_3 \in \mathcal{R}^{2d_e \times d}$ and $d_3 \in \mathcal{R}^d$ are learnable model parameters.

$$e_i^h = Dropout(([e_i^s, \bar{e}_i^f] \cdot W_3 + d_3) + e_i^y), i = 0, 1, \cdots, L-1 \quad (6)$$

Let $e_i^s \in \mathcal{S}$ be the skill ID embedding, and $\bar{e}_i^f$ be the smoothed question difficulty level embedding at position $i$ of the query sequence, $i$=1, 2, $\cdots$, $L$. The query sequence is converted to input vectors to the FFN prediction layer as follows.

$$e_i = Dropout([e_i^s, \bar{e}_i^f] \cdot W_3 + d_3), i = 1, 2, \cdots, L \quad (7)$$

Note that we do not use skill difficulty levels to replace skill IDs because students' knowledge states are assessed in terms of individual skills. Skills with the same difficulty level are not exchangeable.

## 3.3 Question difficulty consistent constraint

Question difficulty consistent constraint (QDCC) requires the predicted scores to be consistent with question difficulty levels. More specifically, given a student, a question with a higher difficulty level should have a lower probability of being answered correctly by the student than a question on the same skill with a lower difficulty level. To impose this constraint, for each question $q$ in the query sequence, we randomly sample a question $q'$ such that $q'$ has the same skill as $q$ to form a sequence of alternative query questions. The alternative question sequence is processed in the same way as the original query sequence as shown in Figure 1. We first pass the alternative question sequence to the embedding layer, concatenate its embeddings with hidden states of LSTM, and then pass the resultant vector to the FFN prediction layer to get predicted scores $\hat{y}'$s on the alternative questions. We then compare $\hat{y}'$ and $diff(q')$ with $\hat{y}$ and $diff(q)$ of the original query question to calculate question difficulty consistent loss $\mathcal{L}_{QDCC}$ as follows.

$$\mathcal{L}_{QDCC} = \frac{1}{L} \sum_{i=1}^{L} |(\hat{y}_i - \hat{y}_i') - (diff(q_i') - diff(q_i))| \quad (8)$$

Given a position $i$, if question $q_i'$ is more difficult than $q_i$, that is, $diff(q_i') > diff(q_i)$, then $\hat{y}_i'$ should be smaller than $\hat{y}_i$. As alternative question sequences pass through only the Embedding layer and the FFN prediction layer, calculating $\hat{y}'$s does not incur much overhead.

## 3.4 Sequence loading

During the training phase, sequences are sampled from students' full learning activity sequences randomly. In each epoch, students with more activities are sampled more frequently. More specifically, the frequency that a student $u$ is sampled in each epoch is calculated as $\lceil N_u/(L+1) \rceil$, where $N_u$ is the number of activities of student $u$

**Table 1: Dataset statistics**

| datasets | #students | #skills | #questions | #activities | % of corrects |
|---|---|---|---|---|---|
| assit09 | 3168 | 150 | 26,628 | 341,879 | 64.5% |
| assit17 | 1708 | 102 | 3,162 | 936,572 | 37.3% |
| assit09 | 571 | 138 | 52,846 | 813,632 | 76.7% |
| assit09 | 1560 | 154 | 125,916 | 2,287,184 | 77.2% |

and $L$ is the length of the sequences to be fed to knowledge tracing models. Once a student is sampled, a random position from this student's full activity sequence is picked as the ending position of the sampled segment. Using this sampling method, for a same student, different segments are sampled from this student's full activity sequence in different epochs, which has some regularization effect on model performance. All the sampled sequences have length $L+1$. Sampled sequences with length less than $L+1$ are padded with zeros at the beginning of the sequences.

During the inference phase, every testing activity $x$ is used as the last activity of a sequence, and the $L$ activities prior to $x$ are used to form a length-($L+1$) testing sequence to be passed to knowledge tracing models.

## 4 A PERFORMANCE STUDY

In this section, we first introduce the datasets and experiment settings used in our performance study, and then present the results of the following experiments: 1) comparing prediction performance of QDCKT with state-of-the-art knowledge tracing models; 2) ablation studies to show the effectiveness of the three techniques used in QDCKT: replacing question IDs with question difficulty levels, combining embeddings of nearby question difficulty levels using a Hann function and question difficulty consistent constraint (QDCC); 3) examining the predictions made by baseline models and variants of QDCKT to see how much they are aligned with question difficulty levels; and 4) comparing running time efficiency and memory consumption of different models.

### 4.1 Experiment settings

We use four datasets in our experiments and their statistics are listed in Table 1. For all the datasets, students with less than 10 activities are removed. The statistics are calculated after the removal. The last column is the percentage of correct answers on the datasets.

- **assist09** [1] was collected on the ASSISTments platform in the school year of 2009-2010. There are three versions of the dataset, and we use the one on skill builder. On this dataset, a question may have more than one skills, and we map the combined skills to one single skill ID.
- **assist17** [2] was also collected on the ASSISTments platform and used in ASSISTments Data Mining Competition 2017. It contains student responses to math questions across two academic years.
- **algebra05** and **algebra06**[39] were used for KDD Cup 2010 Educational Data Mining Challenge. On these two datasets,

---

[1]https://sites.google.com/site/assistmentsdata/home/2009-2010-assistment-data
[2]https://sites.google.com/view/assistmentsdatamining/dataset

**Table 2: Comparison with baseline models.**

| models | assist09 | | assist17 | | algebra05 | | algebra06 | |
|---|---|---|---|---|---|---|---|---|
| | AUC | RMSE | AUC | RMSE | AUC | RMSE | AUC | RMSE |
| DKT | 0.7185±0.0066 | 0.4444±0.0034 | 0.7272±0.0062 | 0.4466±0.0025 | 0.6757±0.0099 | 0.4065±0.0025 | 0.7254±0.0032 | 0.3929±0.0052 |
| DKVMN | 0.7203±0.0105 | 0.4634±0.0068 | 0.7524±0.0057 | 0.4394±0.0027 | 0.7829±0.0031 | 0.3821±0.0041 | 0.8135±0.0022 | 0.3624±0.0037 |
| SAKT | 0.7160±0.0084 | 0.4650±0.0053 | 0.7204±0.0046 | 0.4463±0.0012 | 0.7938±0.0016 | 0.3732±0.0030 | 0.8160±0.0023 | 0.3605±0.0034 |
| AKT | 0.7852±0.0051 | 0.4220±0.0044 | 0.7834±0.0031 | 0.4244±0.0021 | 0.8173±0.0033 | 0.3637±0.0030 | 0.8400±0.0020 | 0.3457±0.0036 |
| LPKT | 0.7539±0.0216 | 0.4474±0.0047 | 0.6687±0.0037 | 0.4911±0.0007 | 0.8059±0.0059 | 0.4002±0.0032 | 0.8265±0.0032 | 0.3940±0.0035 |
| DIMKT | 0.7785±0.0056 | 0.4242±0.0047 | 0.7869±0.0038 | 0.4238±0.0027 | 0.8186±0.0028 | 0.3621±0.0027 | 0.8413±0.0013 | 0.3463±0.0037 |
| QIKT | 0.7482±0.0047 | 0.4432±0.0051 | 0.6509±0.0033 | 0.4969±0.0005 | 0.8051±0.0028 | 0.3749±0.0041 | 0.8245±0.0014 | 0.3663±0.0049 |
| QDCKT | **0.7893**±0.0049 | **0.4164**±0.0038 | **0.7931**±0.0045 | **0.4195**±0.0028 | **0.8221**±0.0026 | **0.3601**±0.0031 | **0.8441**±0.0016 | **0.3451**±0.0041 |

**Table 3: Ablation studies. Number of question difficult levels is set to 1000.**

| models | assist09 | | assist17 | | algebra05 | | algebra06 | |
|---|---|---|---|---|---|---|---|---|
| | AUC | RMSE | AUC | RMSE | AUC | RMSE | AUC | RMSE |
| QDCKT | **0.7893**±0.0049 | **0.4164**±0.0038 | 0.7931±0.0045 | 0.4195±0.0028 | **0.8221**±0.0026 | 0.3601±0.0031 | 0.8441±0.0016 | 0.3451±0.0041 |
| w/o QDCC | 0.7815±0.0055 | 0.4217±0.0043 | 0.7932±0.0040 | 0.4197±0.0025 | 0.8199±0.0023 | 0.3614±0.0032 | 0.8431±0.0019 | 0.3448±0.0034 |
| $l$=1 | 0.7870±0.0053 | 0.4174±0.0040 | 0.7966±0.0043 | 0.4182±0.0029 | 0.8215±0.0025 | **0.3599**±0.0030 | **0.8442**±0.0018 | **0.3441**±0.0037 |
| $l$=1,w/o QDCC | 0.7765±0.0056 | 0.4252±0.0044 | 0.7974±0.0045 | 0.4181±0.0029 | 0.8173±0.0044 | 0.3657±0.0035 | 0.8428±0.0019 | 0.3451±0.0036 |
| ID | 0.7846±0.0053 | 0.4385±0.0065 | **0.8009**±0.0043 | **0.4161**±0.0030 | 0.8143±0.0028 | 0.3704±0.0043 | 0.8381±0.0019 | 0.3527±0.0042 |
| ID,w/o QDCC | 0.7838±0.0041 | 0.4366±0.0083 | 0.7924±0.0040 | 0.4242±0.0025 | 0.8064±0.0053 | 0.4221±0.0145 | 0.8385±0.0024 | 0.3561±0.0037 |

**Table 4: Variants of QDCKT**

| models | $f_q$ | question ID | $l$ for Hann func | QDCC |
|---|---|---|---|---|
| QDCKT | yes | no | 21 | yes |
| w/o QDCC | yes | no | 21 | no |
| $l$=1 | yes | no | 1 | yes |
| $l$=1,w/o QDCC | yes | no | 1 | no |
| ID | no | yes | - | yes |
| ID,w/o QDCC | no | yes | - | no |

values of column "Problem Hierarchy" are used as skills, and combinations of values in "Problem Name" column and "Step Name" column are used as questions. All values are converted to lower case. We also replace concrete numbers in "Step Name" column by variable names like $a$, $b$, $c$ so that similar step names can be merged together and regarded as the same step.

The following baselines are included in our experiments:

- DKT [34]: is the first knowledge tracing algorithm using a deep learning model, and it uses skill IDs and student responses only. Different from the original algorithm in [34] which uses fixed vectors for skills, we use learnable embeddings for skills in our experiments.
- DKVMN [51]: uses a key-value memory network for knowledge tracing. We use question IDs and student responses as its inputs.

- SAKT [29]: uses an attention sublayer plus an FFN sublayer for knowledge tracing. We use question IDs and student responses as its inputs.
- AKT [3] [11]: is an attentive knowledge tracing model with three blocks, and it uses a context-aware distance measure to model forgetting. It takes skill IDs, question IDs and student responses as inputs.
- LPKT [36]: aims to monitor students' knowledge states through consistently modeling their learning process. It takes question IDs, response time (in seconds), elapsed time from previous question (in minutes), question-skill matrix and student responses as inputs. Note that LPKT does not create embeddings for skill IDs.
- DIMKT [4] [35] explicitly uses question difficulty levels and skill difficulty levels together with skill IDs, question IDs and student responses as inputs.
- QIKT [5]: is an interpretable knowledge tracing model combining LSTM with item response theory. It takes skill IDs, question IDs and student responses as inputs.

We implement DKT and SAKT ourselves using PyTorch. For DKVMN, DIMKT and QIKT, we obtain their model implementations from the pyKT library [24]. The implementations of AKT and LPKT are downloaded from the website provided in the original papers. All the models use the same sequence loader as described in Section 3.4 for training and testing.

Our experiments were conducted on a NVIDIA A40 GPU with 48GB memory. The performance of the models is evaluated using AUC (Area Under Curve) and RMSE (Root Mean Square Error). On

---

[3]https://github.com/arghosh/AKT
[4]https://github.com/bigdata-ustc/EduKTM

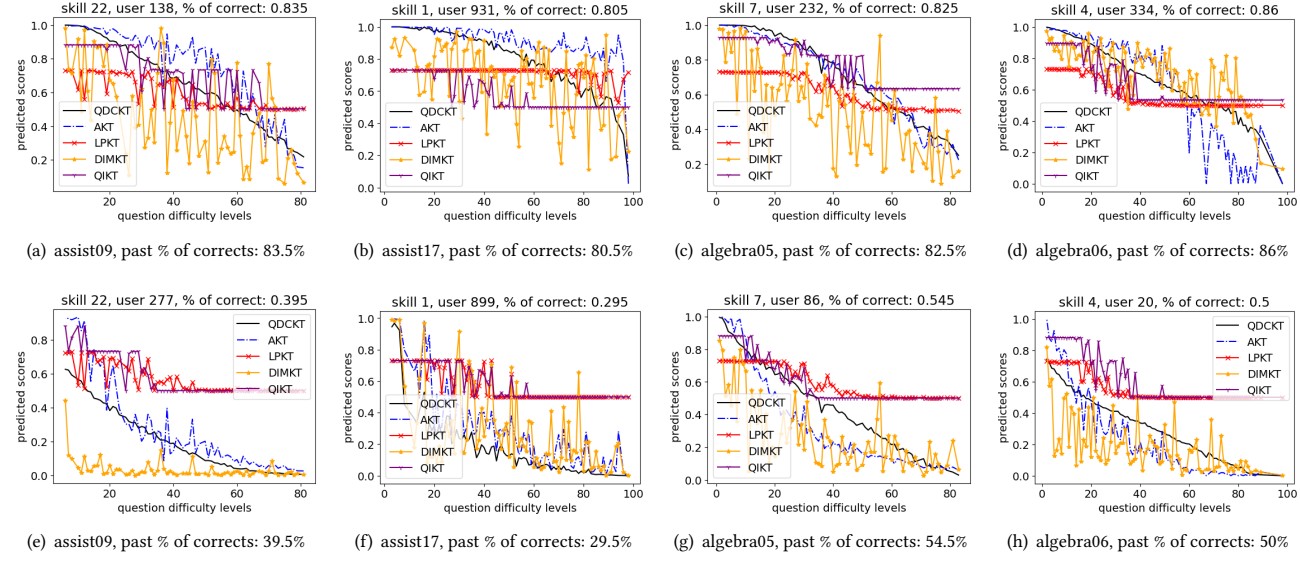

Figure 2: Scores predicted by different models versus question difficulty levels. The first row shows scores predicted on a high-performing student and the second row shows scores predicted on a low-performing student over a same skill. "past % of corrects" is calculated over the selected length-200 history sequence.

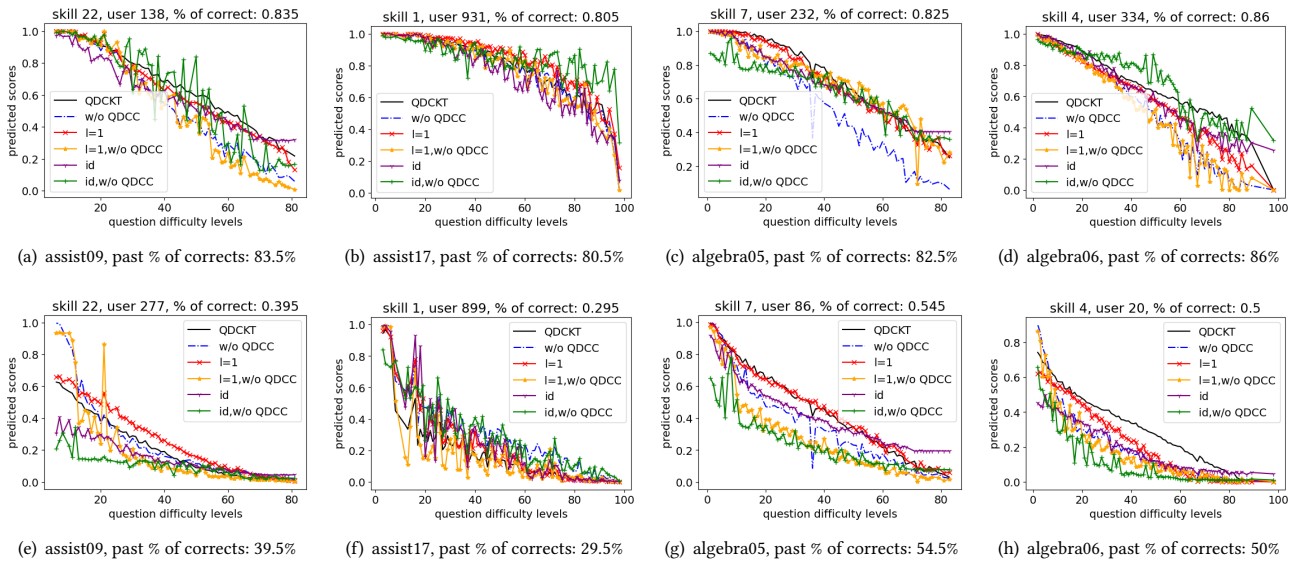

Figure 3: Scores predicted by variants of QDCKT versus question difficulty levels. The first row shows scores predicted on a high-performing student and the second row shows scores predicted on a low-performing student over a same skill.

all datasets, the following hyper-parameters are used: sequence length $L$ is set to 200, batch size is set to 256, embedding dimension and model input dimension are both set to 64, hidden layer dimension of FFN is set to 512, number of RNN and attention layers is set to two, and attention head number is set to 8. Adam optimizer ($\beta_1$=0.9, $\beta_2$=0.999, $\epsilon$=1e-08) is used for model training. All models are trained using one cycle of cosine annealing scheduling with a

minimum learning rate of 0.0001, and the number of epochs is set to 100. Window size $l$ for Hann function is set to 21. Grid search is used to select the best hyper-parameter values on validation data. The maximum learning rate is selected from [0.01, 0.003, 0.001, 0.0003, 0.0001]. Dropout rate is selected from [0, 0.1, 0.2, 0.3, 0.4, 0.5]. Question difficulty level number $N$ is selected from [100, 1000]. Number

of latent concepts for DKVMN is selected from [4, 8, 16, 32, 64]. Five-fold cross validation is used to evaluate model performance.

QIKT model consumes more memory than other models. We get out-of-memory error when running QIKT on *algebra05* and *algebra06* datasets with batch size of 256. We reduce the batch size for QIKT on the two datasets to 128 and 64 respectively. The number of questions on *assist09*, *algebra05* and *algebra06* is very large. To avoid over-parameterization, we use a smaller dimension for question ID embeddings for LPKT and DIMKT. The dimension of question embeddings for these two models is tuned using values from [1, 2, 4, 8, 16, 32, 64].

## 4.2 Comparing with baselines

Table 2 shows the mean and standard deviation of AUC and RMSE of the models evaluated using five-fold cross validation. The best performance is highlighted in **bold**. The second best performance is highlighted using underline. DKT uses only skill IDs and responses, so it has the lowest AUC and highest RMSE. DKVMN and SAKT uses question IDs and student responses. Their performance is much better than DKT, but is worse than other models that use both question information and skill IDs. Our model performs the best among all models with the highest AUC and lowest RMSE on all the four datasets. DIMKT performs the second best and it also uses question difficulty levels.

## 4.3 Ablation studies

In this experiment, we study the impact of the three techniques used in our QDCKT model: using question difficulty levels to replace question IDs, combining embeddings of nearby difficulty levels and the question difficulty consistent constraint described in Section 3.3. Table 4 shows the configurations of several variants of QDCKT. In this table, second column indicates whether question difficulty levels are used, third column indicates whether question ID embeddings are used, fourth column indicates the window size of Hann function for combining embeddings of nearby difficulty levels if applicable, and the last column shows whether the question difficulty consistent constraint is used. When question ID embeddings are used, we also choose the dimension of question ID embeddings from [1, 2, 4, 8, 16, 32, 64] to avoid over-parameterization.

The AUC and RMSE of the variants are shown in Table 3. Using question difficulty levels improves model performance on datasets *assist09*, *algebra05* and *algebra06* where the number of questions is large. On these three datasets, many questions have only a few activities, that is, there are many cold-start questions on these three datasets. For example, nearly 60% questions have less than 10 activities on *assist09*, and this number increases to be more than 70% on *algebra05* and *algebra06*. The question difficulty consistent constraint improves model performance on the same three datasets. The improvement is more obvious when embeddings of nearby difficulty levels are not combined ($l$=1). It also improves model performance significantly when question ID embeddings are used on datasets *assist17* and *algebra05*. Combining embeddings of nearby difficulty levels improves model performance slightly on *assist09*, *algebra05* and *algebra06*.

On dataset *assist17*, the three techniques do not seem to be helpful. However, question difficulty consistent constraint is still able to improve model performance significantly when question ID embeddings are used. Using question difficulty levels to replace question IDs decreases model performance slightly when QDCC is already in use. It still improves model performance when QDCC is not in use. QDCKT has the largest performance gain over baseline models on this dataset.

## 4.4 Score consistency

In this experiment, we plot the predicted scores produced by different models versus question difficulty levels. We randomly sample a high-performing student and a low-performing student from the testing data on each dataset, and select a skill that has many questions and these questions are at various difficulty levels. For the selected skill, one question is selected at each question difficulty level. For each sampled student, we randomly select a length-200 segment from her/his learning activity sequence as the history sequence, and then use the selected skill and its selected questions as the next question whose class label is to be predicted. Figure 2 shows the scores generated by different models (y-axis) versus the question difficulty levels of the selected questions (x-axis). The predictions made by our QDCKT model are very well aligned with question difficulty levels. They are also aligned with students' historical performance. The high-performing student has higher predicted scores than the low-performing students on a same skill at a same difficulty level.

The predictions made by other models such as DIMKT can fluctuate dramatically despite the fact that DIMKT uses both question difficulty levels and skill difficulty levels. This is not desirable as end users may find it hard to use and trust such predictions. LPKT and QIKT often produce flat predictions, that is, the predicted scores do not change when question difficulty changes. This may be fine if a student does not know a skill hence the predicted score at any difficulty level is close to 0, or a student already masters a skill very well hence the predicted score at any difficulty level is close to 1. However, the scores predicted by the two models are flat mostly at around 0.5, which means the models are uncertain whether the student masters the skill or not.

Figure 3 shows the scores produced by variants of QDCKT versus the question difficulty levels of the selected questions. Even without using the three techniques, our model shows better alignment with question difficulty levels than baseline models. Among the variants of QDCKT, "ID,w/o QDCC" has the biggest fluctuation, followed by "$l$=1,w/o QDCC". Using QDCC ("ID" and "$l$=1" ) can effectively smooth predicted scores. So does combining nearby embeddings of question difficulty levels ("w/o QDCC" which uses a window size of 21 for Hann function).

## 4.5 Running time and memory consumption

Table 5 shows the mean running time of one epoch of different models. Table 6 shows the maximal memory usage of different models with a batch size of 256 during training. For QIKT, we have to set batch size to a smaller value of 128 and 64 on *algebra05* and *algbrea06* respectively because of out-of-memory error when batch size is larger. DIMKT consumes the least amount of memory, but it is more than 10 times slower than QDCKT. QDCKT consumes

**Table 5: Mean running time of one epoch of different models (seconds).**

| models | assist09 | assist17 | algebra05 | algebra06 |
|---|---|---|---|---|
| DKT | **0.8** | **1.0** | **0.8** | **1.4** |
| DKVMN | 3.9 | 5.2 | 5.1 | 10.8 |
| SAKT | 1.1 | 1.5 | 1.4 | 2.5 |
| AKT | 6.8 | 9.5 | 7.5 | 19.7 |
| LPKT | 19.6 | 25.2 | 21.6 | 57.5 |
| DIMKT | 11.9 | 16.7 | 13.1 | 37.6 |
| QIKT | 4.4 | 2.0 | 8.0 | 51.5 |
| QDCKT | 1.4 | 1.5 | 1.6 | 2.7 |
| w/o QDCC | 1.1 | 1.2 | 1.2 | 1.9 |
| $l$=1 | 1.2 | 1.4 | 1.2 | 2.3 |
| $l$=1,w/o QDCC | 0.9 | 1.2 | 1.1 | 1.9 |
| ID | 1.2 | 1.4 | 1.2 | 2.3 |
| ID,w/o QDCC | 0.9 | 1.2 | 1.1 | 1.9 |

**Table 6: Maximal memory consumption of different models during training (GB)**

| models | assist09 | assist17 | algebra05 | algebra06 |
|---|---|---|---|---|
| DKT | 2.0GB | 1.9GB | 2.0GB | 2.0GB |
| DKVMN | 4.0GB | 2.4GB | 2.4GB | 2.3GB |
| SAKT | 2.6GB | 2.6GB | 3.0GB | 3.0GB |
| AKT | 17.0GB | 17.0GB | 17.0GB | 17.0GB |
| LPKT | 11.9GB | 8.1GB | 11.1GB | 12.0GB |
| DIMKT | **1.5GB** | **1.5GB** | **1.5GB** | **1.5GB** |
| QIKT | 33.2GB | 4.4GB | 22.3 | 38.4GB |
| QDCKT | 2.3GB | 2.4GB | 2.4GB | 2.4GB |

slightly more memory than DIMKT. All variants of QDCKT consumes similar amount of memory ranging from 2.1GB to 2.5GB. DKT is the fastest model among all models, and its memory usage is also small. However, its performance is also the worst among all the models. QDCKT makes a much better trade-off among running time, memory usage and prediction performance.

## 5 SUMMARY AND CONCLUSION

In this paper, we propose a model which uses question difficulty levels to replace question ids for knowledge tracing and adopts two techniques to further smooth the predictions. It shows better performance than several latest knowledge tracing models in term of prediction accuracy, prediction quality and running efficiency. We use a LSTM sublayer to generate representations of historical sequences and a FFN as the prediction layer. They can be replaced by other deep learning units. For example, the LSTM sublayer can be replaced by a Transformer Encoder, and the FFN prediction layer can be replaced by a multi-head attention sublayer. We have explored this model architecture and it has similar prediction performance with the current architecture, but is slower.

The predictions made by our model are more consistent with question difficulty levels and can be more readily used to estimate knowledge states of students over skills than existing models that rely on question IDs. This brings us one step closer to more interpretable and more trustworthy knowledge tracing models. As our future work, we will explore how to make our model more explainable.

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
