# OpenReview forum: "Question Difficulty Consistent Knowledge Tracing"
_ACM.org/TheWebConf/2024/Conference — TheWebConf24_

### Official Review · Reviewer_XXmg · 2023-11-22

**Novelty:** 4
**Technical Quality:** 4

**Review:**

This paper proposed a novel method for knowledge tracing, called Question Difficulty Consistent Knowledge Tracing. It aims to calculate difficulty levels for questions answered in historical data of student’s record, and utilize a LSTM based network to predict on the outcome, given any arbitrary difficulty levels. This method is different from existing question-based work that predict on whether the student can correctly answer a new question. Experiments on the model’s performance against baselines are conducted, as well as ablation studies, score consistency and running efficiency. However, the author did not provide a convincing explanation for why the new methods can be beneficial and are conceptually superior than the existing work, since some of the claims in the introduction section are not grounded by experiments. Thus, although the model has achieved the best performance against the baselines, the paper either requires revision on the work’s objective, or more experiments are needed.

**Questions:**

1. The author state that the goal of this model is to “predict whether a student can answer ANY question of a given skill at a given difficulty”, which is different from existing methods that aims to “predict whether a student can answer the next specific question of a given skill correctly”. It seems not very reasonable to replace the “old” goal with the proposed new goal, since they are not mutually exclusive, and this new model can just add question ID as a feature and become capable of directly predicting label for a given question ID. The goal can be changed to “predict whether a student can answer a specific question or any question of specific skill and difficulty level”, which gives the model predictive capability with 2 forms of input: question ID or difficulty.

2. Continuing from the previous question: Can the author provide more explanation about the reason for excluding question ID from the model? From my understanding, the existing work directly uses question ID as a feature, while the proposed work utilizes difficult level of q (calculated by Eq. 4 and 5). Why not utilize them both in the model?

3. From the difficulty equation, which is Eq. 4, we can see that for a given question q_i, the diff(q) is calculated by a deterministic function. Then for a given dataset D, all q_i are predetermined, so f_q works like an extra calculated feature column. Intuitively, using

both difficulty and question ID as input feature should be included as one of the experiment settings, but in the Table 3 and 4, there is no experiment on QDCKT+ID. Thus, the experiments conducted is somewhat incomplete.

4. Can author include more statistics of dataset in Table 1? Such as average student question, skills, etc.

5. For section 4.4, can author add more explanations for the plot? For example, what would the predicted line looks like if the model has a perfect fit?

6. In introduction, some of the statements are not grounded by experiments. For example, in the third paragraph, the author stated: “Furthermore, by using question difficulty levels to replace question IDs, we can also alleviate the cold-start problem in knowledge tracing as online learning platforms are updated frequently with new questions”. But there are no experiments that directly demonstrate how this model alleviate the cold-start problem. And in the Experiments Settings, “students with less than 10 activities are removed”. It might be very helpful to use those removed data to demonstrate the potential benefit of the proposed model on the cold-start problem.

**Reviewer Confidence:**

4: The reviewer is certain that the evaluation is correct and very familiar with the relevant literature

**Scope:**

4: The work is relevant to the Web and to the track, and is of broad interest to the community

---

### Official Review · Reviewer_Hoox · 2023-11-29

**Novelty:** 6
**Technical Quality:** 6

**Review:**

This work proposes a novel approach to knowledge tracing which is a problem that tracks student knowledge by utilizing the students' responses to questions. This paper takes a step back and, instead of predicting whether a student will successfully complete the next question, it predicts whether the student will complete any question right at a specific difficulty level. The method is evaluated against other competing approaches in terms of AUC scores, RMSE, consistency, runtime, and memory consumption. An ablation study is also performed to evaluate the contribution of each component of the model.

The paper is easy to follow and read. The notions are explained properly, as well as the novel contributions of this paper. The experimental evaluation is extensive. The datasets used are publicly available. Most importantly, the strongest contribution of this paper is its ability to perform well related to the cold start problem, i.e., when a new question becomes available. The previous approaches would not be able to generate good predictions for these cases, as the new questions do not appear in many (or any) previous sequences. The proposed approach is properly motivated and its components are intuitive.

While most of the paper is clear, the description of the proposed model is not as well explained. Are all the elements s_q, e_q^h, h_q simple numbers (like f_q) The paper does not discuss well (nor at least offer some references) what is the s_0, and how exactly the embedding layer works, the matrices S, D, and C. Why do the elements [1, L-1] appear both in history and query sequences? Additionally, the hyperparameter alpha from Eq. 2 is not discussed (e.g., values used in experiments).

Suggestions:
- since the paper tackles the cold-start problem, there could be some evaluation of how it performs on questions with varying popularity.
- The statistics in tables 5 and 6 could be averaged out over all the datasets to save some space since we do not have a lot of other dataset-specific info to understand how the running time and memory consumption relate to that.
- Figures 2 and 3 are too complicated to be easily understood; only the best 3-4 methods could be used.

**Questions:**

- Why do the elements [1, L-1] appear both in history and query sequences?
- Why is there a claim about the interpretability of the model in the conclusion?

**Reviewer Confidence:**

3: The reviewer is confident but not certain that the evaluation is correct

**Scope:**

3: The work is somewhat relevant to the Web and to the track, and is of narrow interest to a sub-community

---

### Official Review · Reviewer_Npp2 · 2023-11-29

**Novelty:** 5
**Technical Quality:** 5

**Review:**

This paper proposes a new knowledge tracing model where question difficulty levels instead of question IDs (in the state-of-the-art methods) are incorporated to predict a redefined problem from whether a student can answer a specific question correctly to whether a student can answer any question of a given skill at a given difficulty level. This proposed solution not only provides a more direct translation to students' knowledge states over skills but also helps mitigate the cold-start problem in knowledge tracing. The framework also integrates two smoothing methods to further facilitate the prediction.

The question of this paper is well defined and highly relevant to the data mining field, especially given the background that online education has increasing demand after the pandemic. In general, the paper was well written with extensive experiments on baselines and ablation studies. I specifically have two questions, the answer for which hopefully help the paper development.

*Potential data leakage
Based on the sampling strategy described in the paper, for each data split the student and the progress state were randomly drawn and one student could appear multiple times in different data splits. Therefore, it's likely that the training split 1, student A's status at time T2 is collected, and the testing split contains student A's status at time T1 (T1<T2).  However, in this case, the model may already "see" student A's question-answer result at T1 in the training phase. More discussion on this would be expected.

* Performance
It seems like the proposed model only moderately outperforms the best baseline (such as 0.7893 vs 0.7852). There are two questions related to performance given that: 1. How did you set up the experiment for baselines? Did you use the same grid search? 2. How to justify the outperformance on such a small scale is important?

**Questions:**

Clarifications on 1. potential data leakage; 2. model performance and baseline experiments

**Reviewer Confidence:**

3: The reviewer is confident but not certain that the evaluation is correct

**Scope:**

3: The work is somewhat relevant to the Web and to the track, and is of narrow interest to a sub-community

---

### Official Review · Reviewer_pJyo · 2023-11-30

**Novelty:** 4
**Technical Quality:** 6

**Review:**

In this paper, the authors propose a novel approach to knowledge tracing, replacing question IDs with difficulty levels in deep learning models. This shift allows predictions to focus on a student's ability to answer any question of a specific skill at a given difficulty level, facilitating cold start. The authors introduce two techniques to enhance predictions, involving difficulty level embeddings and a consistent constraint on predicted scores. Experiments show the approach, combining LSTM for learning representations and a feed-forward neural network for predictions, outperforms more complex models in terms of consistency with question difficulty levels, efficiency, and accuracy.

Strengths
+ Very well-written and well-structured, the authors conveyed their idea in a clear and concise way.
+ Well-summarized contributions and connection to prior work, with well-motivated choices.
+ The paper covers four datasets, from two different platforms, which allow to have a good level of generalizability.

Limitations
- Knowledge tracing is a very important topic for the community working on educational data mining. As the paper stands, I found however hard to grasp the core connection with respect to the topics of the conference. The connection seems limited, as the authors mainly touch on the Web for the data collection. Without a proper contextualization on how the proposed method affect the Web, the paper can be only of marginal interest for this community and, in any case, such interest might be narrowed to a limited sub-community which is working on educational data mining.
- The novel contribution of this paper appears to be an incremental addition to existing knowledge about the task. In this sense, I appreciated the extensive results the authors provided to counterbalance the limited technical novelty. Nevertheless I believe that the gains showed with respect to the best baseline, e.g., in Table 2, seem to tell that the proposed contribution requires more elaboration to have concrete impact on students' learning.
- The experimental results highlight that framework's outperforms of several baselines. However, the reported gains, often at the second or third decimal, leave room for ambiguity regarding the real impact on students' learning. Moreover, the authors merely focus on machine-learning oriented metrics, like AUC and RMSE, without any connection to how this can enable a better learning on the Web. To show the significance of the improvement, it is suggested that the authors consider an online evaluation with A/B testing to provide a more practical context for their findings.
- The results (see Section 4) cover a very large set of experiments and metrics, touching both on accuracy and efficiency in terms of memory and running time. However, from the way they are presented, it is hard to grasp the concrete trade-off of each method with respect to this set of metrics and, this, deciding which method should be preferred with respect to the others once we should move to real-world implementation. Radar plots might be a more effective way of showing the results and let the reader grasp the trade-off easily.
- Although the paper covers various datasets and baselines with detailed implementation information, the absence of shared source code may pose challenges for other researchers attempting to reproduce the work, given also that some baselines were re-created from scratch. Sharing the source code would contribute to the reproducibility.

In conclusion, while the paper makes strides in knowledge tracing, its limited connection to broader conference themes, incremental contribution, and the need for further elaboration on practical impact suggest areas for improvement. Addressing the contextualization of the proposed method's impact on the web, enhancing technical novelty, providing clearer real-world implications through A/B testing, employing more effective result visualization methods, and sharing source code for reproducibility would significantly strengthen the overall contribution of this work.

**Questions:**

- To what extent the gains in the offline experiments might concretely translate into benefits in online experiments with learners?

**Reviewer Confidence:**

4: The reviewer is certain that the evaluation is correct and very familiar with the relevant literature

**Scope:**

3: The work is somewhat relevant to the Web and to the track, and is of narrow interest to a sub-community

---

### Official Review · Reviewer_3cSy · 2023-12-01

**Novelty:** 3
**Technical Quality:** 3

**Review:**

This paper studies the knowledge tracing problem which estimates knowledge states of students based on their historical learning activities.  It is an interesting and practical problem in the education area.  My major concerns are listed as follows:
Cons:
1. The technique novelty is limited. Using question difficulty levels to replace question IDs is intuitive. The improvement is small and incremental.
2.  The most simple way to evaluate the knowledge states of students is to calculate the statistics of different difficulty levels of questions. Such statistical methods should be compared.
3. It's not clear how to get the difficulty level of questions.

**Questions:**

1. How to get the difficulty levels of questions?
2. What's the influence on results by conducting smoothing operations on the embeddings?

**Reviewer Confidence:**

3: The reviewer is confident but not certain that the evaluation is correct

**Scope:**

3: The work is somewhat relevant to the Web and to the track, and is of narrow interest to a sub-community

---

### Decision · Program_Chairs · 2024-01-22

**Decision:**

Accept

**Comment:**

The paper advocates a new principled problem formulation for knowledge tracing. Instead of predicting at the question-ID level, the authors argue for a coarser granularity of the question-difficulty level. Such coarse granularity can be achieved through smoothing and regularization, and slightly outperform SoTA models while ensuring consistency with question difficulty levels.

 Reviewers noted that although the presented methodology is not the most novel, but the problem and the solution are both well-motivated and important to the area. The reviewers also indicate that their questions are mostly addressed during the rebuttal. I encourage the authors to carefully revise their paper according to the reviews and rebuttal comments.